# Harnessing gut microbes for glycan detection and quantification

Jennifer L. Modesto[1,2,3], Victoria H. Pearce [1,2,3] & Guy E. Townsend II [1,2,3] ✉

Glycans facilitate critical biological functions and control the mammalian gut microbiota composition by supplying differentially accessible nutrients to distinct microbial subsets. Therefore, identifying unique glycan substrates that support defined microbial populations could inform therapeutic avenues to treat diseases via modulation of the gut microbiota composition and metabolism. However, examining heterogeneous glycan mixtures for individual microbial substrates is hindered by glycan structural complexity and diversity, which presents substantial challenges to glycomics approaches. Fortuitously, gut microbes encode specialized sensor proteins that recognize unique glycan structures and in-turn activate predictable, specific, and dynamic transcriptional responses. Here, we harness this microbial machinery to indicate the presence and abundance of compositionally similar, yet structurally distinct glycans, using a transcriptional reporter we develop. We implement these tools to examine glycan mixtures, isolate target molecules for downstream characterization, and quantify the recovered products. We assert that this toolkit could dramatically enhance our understanding of the mammalian intestinal environment and identify host-microbial interactions critical for human health.

The mammalian gut microbiota is a critical human health factor and a premier target for therapeutic modulation. The gut microbiota composition is influenced by the host diet, intestinal mucosa, and co-resident microbes, which all contribute to the availability of complex oligo- and polysaccharides, collectively referred to herein as glycans[1–5]. Glycans are highly abundant macromolecules found in every domain of life, mediate critical cellular processes, and exhibit extraordinary structural diversity conferred by unique non-linear arrangements, glycosidic linkages, and monosaccharide components[6,7]. Altering glycan availability in the intestine modulates the gut microbiota by favoring the expansion of microbial populations that can consume individual glycan structures, which typically requires differentially encoded enzymatic activities[8–11]. The mammalian intestine can exhibit disease-associated alterations in mucosal glycosylation potentially influencing the abundance of gut microbial populations[12,13]. Thus, evaluating glycans associated with intestinal diseases could reveal

disease biomarkers, therapeutic targets, or prebiotic compounds. However, unlike nucleic acids and proteins, glycan structural complexity limits high-throughput examinations of heterogeneous glycan mixtures using individual methodologies and frequently requires parallel tandem approaches. Therefore, tools and methods are necessary to efficiently survey mixtures for unique glycan structures that could modulate microbial abundance and activity in the gut.

Microbes have evolved systems dedicated to glycan recognition and consumption from a myriad of biological sources[14]. Members of the gut microbiota encode vast repertoires of glycan utilization machinery conferring unparalleled access to the diverse substrates present in the intestinal lumen facilitating gut colonization[4,9–11]. For example, a dominant and abundant bacterial group present in the mammalian gut, called the *Bacteroides*, encode discreet, tightly linked, coordinately regulated gene clusters called polysaccharide utilization loci (PULs) that mediate the consumption of structurally distinct

[1]Department of Biochemistry & Molecular Biology, Penn State College of Medicine, Hershey, PA 17033, USA. [2]Penn State Microbiome Center, The Pennsylvania State University, State College, PA 16802, USA. [3]Center for Molecular Carcinogenesis and Toxicology, The Pennsylvania State University, State College, PA 16802, USA. ✉e-mail: gtownsend@pennstatehealth.psu.edu

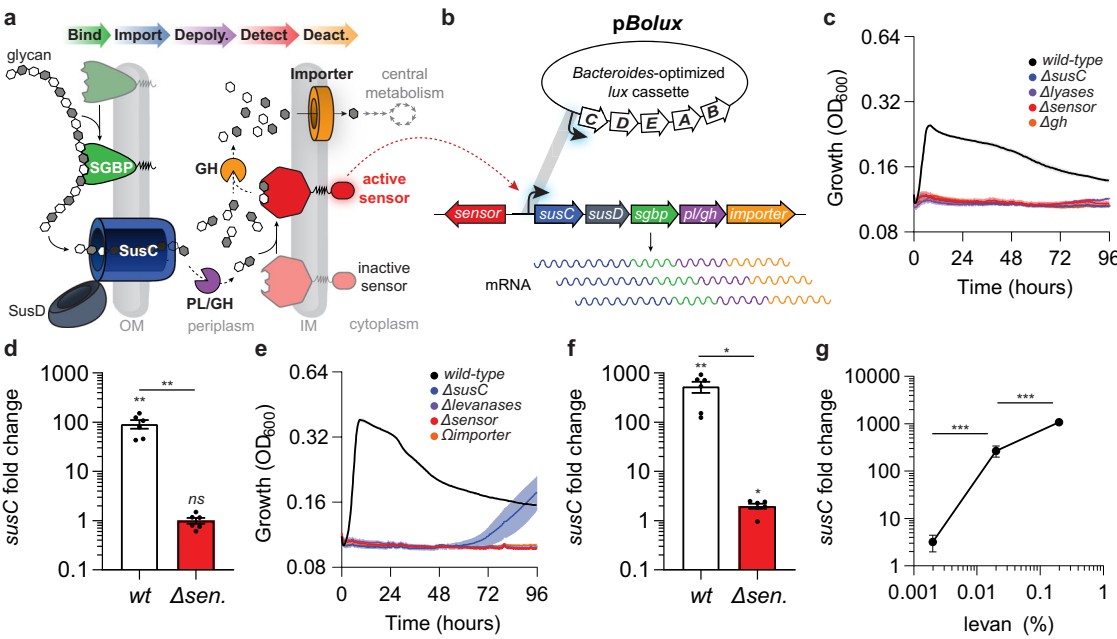

**Fig. 1 | Bt PUL transcription reflects target glycan abundance. a** A cartoon depicting prototypical glycan utilization by PUL-encoded gene products. SGBPs (green) specifically bind target glycans in the extracellular milieu, which are transported across the outer membrane (OM) by the SusCD complex (blue). Internalized glycans are depolymerized by glycosyl hydrolase (GH) and poly-saccharide lyase (PL) activities (purple) generating ligands that convert PUL sensors (red) to their active form. PUL sensors are deactivated following removal of their ligands from the periplasm by GH activities or importation across the inner membrane (IM; orange). **b** Ligand-bound sensors direct increased PUL transcription by sequence-specific interactions with PUL promoters. These promoters can be introduced into p*Bolux* to drive glycan-responsive bioluminescence. **c** Growth of *wild-type Bt* (black) or strains lacking the CS-inducible *susC* (Δ*BT3332*, blue), 3 CS-specific lyases (Δ*BT3324* Δ*BT3350* Ω*BT4410*, purple), CS-sensor (Δ*BT3334*, red), or a glucuronyl hydrolase (Δ*BT3348*, orange) were measured during anaerobic culture in minimal media containing 0.1% CS as a sole carbon source. Values are the mean of 8 biological replicates and error bars are SEM in color-matched shading. **d** The fold increase of the CS-inducible *susC* (*BT3332*) mRNA levels were measured by qPCR in either *wild-type Bt* (open bar) or a strain lacking the CS-sensor (red bar) following introduction of 0.2% CS into the culture media. The fold increase was calculated as the change in transcript levels from cultures before and 2 h after the introduction of

CS. Values are the mean of 6 biological replicates, error bars are SEM, and P-values were computed using a two-tailed student's *t*-test and ** indicates values <0.01 and ns > 0.05. **e** Growth of *wild-type Bt* (black) or strains lacking the levan-inducible *susC* (Δ*BT1763*, blue), 4 levanases (Δ*BT1760-1759* Δ*BT3082* Ω*BT1765*, purple), fructan sensor (Δ*BT1754*, red), or a putative inner-membrane fructose importer (Ω*BT1758*, orange) were measured during anaerobic culture in minimal media containing levan as a sole carbon source. Values are the mean of 8 biological replicates, error bars are SEM in color-matched shading. **f** The fold increase of levan-inducible *susC* (*BT1763*) mRNA levels were measured in either *wild-type Bt* (open bars) or a strain lacking the levan sensor (red bars) following the introduction of 0.2% levan in the culture media. The fold change was calculated as the change in transcript levels from cultures before and 2 h after levan introduction. Values are the mean of 6 biological replicates, error bars are SEM, and P-values were computed using a two-tailed student's t-test and ** indicates values <0.01, * <0.05 and ns > 0.05. **g** The fold change in *BT1763* mRNA levels were measured by qPCR in *wild-type Bt* as described in (**d**), 120 min following the introduction of mixtures of 0.2%, 0.02%, or 0.002% levan supplemented with galactose to 0.5% total carbohydrate. Values are the mean of 6 independent measurements, error bars are SEM, and P-values were calculated by 2-way ANOVA with Tukey's honest significance test and *** represents values <0.001. Source data are provided as a Source Data file.

glycans[9–11]. Many *Bacteroides* species encode vast PUL libraries endowing these organisms with the ability to consume structurally diverse complex polysaccharides derived from the host diet, co-resident microbes, and the intestinal mucosa to successfully thrive in the mammalian gut[10,11,15]. Most PULs are transcriptionally regulated such that individual genes are expressed at low constitutive levels until encountering their target glycans (Fig. 1a), resulting in rapid and dramatic increases in corresponding PUL transcripts (Fig. 1b)[9,11,16–18]. These responses are directed by sensor proteins that recognize mono-, di-, or oligo-saccharide signatures of their corresponding target glycan(s) and subsequently increase PUL transcript levels in a rapid and dramatic fashion (Fig. 1a, b)[11,16,18,19]. The ability of PUL sensors to direct PUL transcription changes in response to glycan availability potentiates their use as biosensors that could be employed to surveille heterogenous mixtures.

Additionally, PULs encode proteins possessing glycan binding, transport, and depolymerization activities that exhibit substrate specificity and confer species-specific access to discreet glycan subsets (Fig. 1a)[3,20,21]. For example, *Bacteroides thetaiotaomicron* (*Bt*) and a closely related species, *Bacteroides ovatus* (*Bo*), both encode similar polyfructan-specific PULs that confer discreet target glycan specificities[3,19]. The *Bt* fructan PUL facilitates the consumption of the β-

2,6-linked polyfructan, levan, which is inaccessible to *Bo*, while the *Bo* fructan PUL facilitates utilization of the β-2,1-linked polyfructan, inulin, which is inaccessible to *Bt*[3,19]. Species-specific polyfructan access is mediated independently of the fructan PUL sensor, which activates PUL transcription in both *Bt* and *Bo* following detection of monomeric fructose[3,19]. Thus, differences in target glycan utilization can rely on PUL-encoded factors that distinguish between compositionally similar yet structurally distinct glycans. The specificity exhibited by these factors, such as surface glycan-binding proteins (SGBPs; Fig. 1a), potentiates their use as reagents to isolate PUL-target glycans from complex heterogenous mixtures aiding in their structural characterization[19,20]. However, the paucity of specific, sensitive, and cost-effective glycan detection methods has hindered the development of such approaches.

We hypothesized that *Bacteroides* PUL sensors could be harnessed as tools to detect their corresponding target glycans by examining changes in PUL transcription following the introduction of various biological samples. However, examining PUL transcription in response to glycan mixtures has relied on molecular approaches such as qPCR, microarrays, and RNAseq to detect transcriptional changes in *Bacteroides* species[9,17,19,22]. Furthermore, the anaerobic growth requirements of *Bacteroides* species have previously limited the

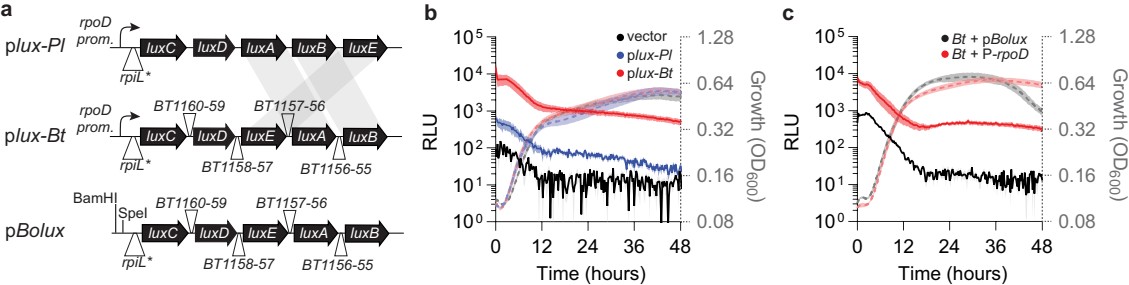

**Fig. 2 | Construction of a Bacteroides-optimized bioluminescent reporter.**
**a** Schematic depicting the construction of a bioluminescent reporter that encodes the entire *Pl* lux cassette under control of the *Bt rpoD* promoter and rpiL* RBS (top); a Bacteroides-optimized lux cassette with rearranged *luxA-E* (indicated by the shaded regions) and *Bt* intergenic regions from a constitutively expressed *Bt* operon (BT1160-1155; middle); or p*Bolux* which has BamHI and SpeI sites positioned upstream of the Bacteroides-optimized lux cassette (bottom) in the multi-copy plasmid pLYL01. **b** Relative luminescence (solid lines) or growth (dashed lines) from *Bt* strains harboring an empty vector (black) or plasmids containing either the *lux* operon from *Pl* (p*lux-Pl*, blue) or the Bacteroides-optimized *lux* cassette (p*lux-Bt*, red) expressed from the *Bt rpoD* promoter and rpiL* RBS were measured during growth in minimal media containing 0.5% galactose as the sole carbon source. **c** The relative luminescence (solid lines) or growth (dashed lines) of *Bt* strains harboring empty p*Bolux* (black) or a plasmid with the *Bt rpoD* promoter cloned into the BamHI and SpeI sites (red) during growth in galactose as the sole carbon source. All values in **b**, **c** are the mean of 8 biological replicates and error bars are SEM in color-matched shading. Source data are provided as a Source Data file.

implementation of traditional transcriptional reporters such as GFP and luciferase to ex vivo aerobic measurements at discreet times[23–26].

Here, we generate a *Bacteroides*-optimized luciferase reporter, p*Bolux*, that accurately reflects transcription during growth in anaerobic culture and is readily quantifiable. Introduction of PUL promoters into p*Bolux* reports PUL promoter activity that sensitively and specifically indicates the presence of target glycans. We demonstrate that these responses require PUL-encoded transport, degradation, and detection machinery in multiple *Bacteroides* species. Moreover, we determine that target glycans elicit dose-dependent responses from strains harboring their corresponding PUL reporters that can be used to quantify glycan abundance as accurately as commercially available kits. Finally, we develop an affinity purification strategy using PUL-encoded SGBPs to isolate individual target glycans from heterogeneous mixtures by employing PUL reporters as glycan detection and quantification tools. This microbial glycomics toolkit is readily scalable and will enable efficient characterization of unknown PUL ligands, identify previously concealed genetic determinants governing microbial glycan utilization, and ultimately reveal the glycomic interface between mammals and their gut microbiotas.

## Results

### Bacteroides PUL sensors promote dose-dependent transcriptional responses to target glycans

*Bacteroides* species frequently encode many PULs (*Bt* and *Bo* encode 88 and 112, respectively)[9,11]), each putatively dedicated to the utilization of distinct glycan substrates. PULs include a *susCD*-like gene pair that encodes a complex necessary for the translocation of a target glycan across the outer membrane (Fig. 1a)[3,9,11,15,27] and the levels of *susCD*-containing transcripts increase rapidly and dramatically following exposure to their cognate target glycan (Fig. 1b)[16,18,19]. For example, transcripts corresponding to the *susC* gene, *BT3332*, located within a *Bt* PUL required for chondroitin sulfate (CS) utilization (Fig. 1c) increase 92-fold after 120 min following the addition of CS to the media (Fig. 1d)[16]. Similarly, transcripts corresponding to another *susC* gene, *BT1763*, located within the *Bt* PUL necessary for levan utilization (Fig. 1e)[19], increase 530-fold following the addition of levan (Fig. 1f)[19]. These increases in *susC* transcript levels require sensor proteins that often bind target glycan-derived ligands in the periplasm and subsequently direct transcriptional responses in the cytoplasm[16,19] (Fig. 1a). Accordingly, *Bt* mutants lacking either the CS or levan PUL sensors, are unable to grow on their corresponding target glycans as sole carbon sources (Fig. 1c, e, respectively) or increase PUL transcription (Fig. 1d, f,

respectively)[16,19], but exhibit no growth defects on galactose (Supplementary Fig. 1a and b, respectively).

Interestingly, *BT1763* and *BT3332* transcript levels exhibited corresponding decreases when *Bt* was supplied tenfold dilutions of levan or CS (Fig. 1g and Supplementary Fig. 2a, respectively) indicating that *Bacteroides* PUL transcriptional responses are concentration-dependent[28]. However, the relationship between *susC* transcripts and glycan concentrations detected at 120 min was not apparent after 60 min following glycan addition (Supplementary Fig. 2b, c), demonstrating the dynamic nature of PUL transcription[16,17,22]. These results suggest that PUL sensors promote dose-dependent transcriptional responses over time, which could be harnessed to report unknown target glycan abundances by measuring changes in corresponding *susC*-transcript levels.

### Construction of a Bacteroides-optimized bioluminescent transcriptional reporter

Quantifying *susC* transcription by qPCR or transcriptomics is expensive and inefficient, requiring kinetic sampling to accommodate variations in PUL transcriptional responses[16,17,22]. Therefore, we sought to engineer a transcriptional reporter that accurately reflects transcript levels in *Bacteroides* species over time during anaerobic growth without additional oxygenation[29]. Previously constructed transcriptional reporters in *Bt* require terminal measurements following cell lysis limiting their applications for kinetic measurements[25,26,30] and fluorophores such as GFP require exposure to oxygen for proper maturation or exhibit high background fluorescence[24,29,31]. In contrast, the production of bacterial LuxCDABE proteins and biosynthesis of the luciferase long-chain aldehyde substrate can be achieved anaerobically although luciferase activity is oxygen dependent[32,33]. We hypothesized that *lux*-mediated bioluminescence could function in *Bacteroides* species during growth under anaerobic conditions when expressed from *Bacteroides* promoters and ribosomal binding sites, which differ from those typically found in *Firmicutes* and *Proteobacteria*[23]. Therefore, we engineered the *Photorhabdus luminescens* (*Pl*) *luxCDABE* cassette in the multi-copy plasmid, pLYL01[17], preceded by the *Bt rpoD* promoter (*BT1311*) and an optimized *Bacteroides* ribosome binding site, rpiL*, that facilitates high levels of gene expression (Fig. 2a)[25,26]. A *wild-type Bt* strain harboring the *Pl* lux cassette exhibited detectable luminescence during anaerobic growth in minimal media containing galactose (Fig. 2b) or glucose (Supplementary Fig. 3a) and grew similarly to a strain harboring the empty vector under identical conditions (Fig. 2b and Supplementary Fig. 3a, respectively).

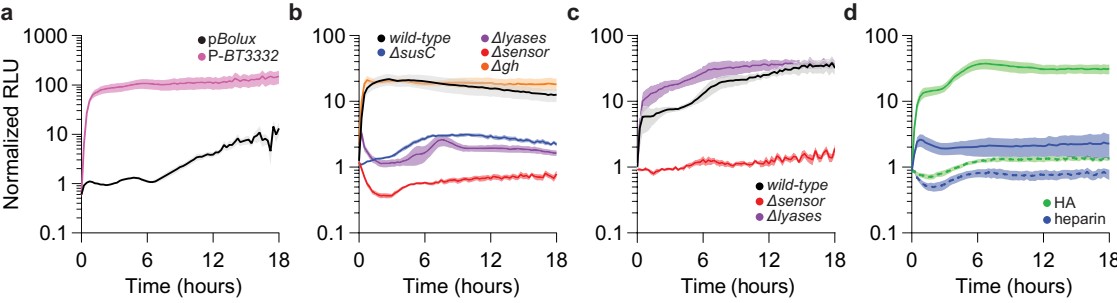

**Fig. 3 | Construction of a glycan-responsive reporter in Bt. a** Relative luminescence from *wild-type Bt* strains harboring p*Bolux* (black) or a plasmid including the promoter region preceding the CS-inducible *susC* gene (P-*BT3332*, pink) following the introduction of CS as the sole carbon source normalized by the relative luminescence of identical cultures supplied galactose. Values are the mean of 12 biological replicates and error is SEM in color-matched shading. **b** Relative luminescence from *wild-type Bt* (black) or strains lacking a CS-inducible *susC* (*ΔBT3332*, blue), 3 CS-specific lyases (*ΔBT3324 ΔBT3350 ΩBT4410*, purple), CS-sensor (*ΔBT3334*, red), or a glucuronyl hydrolase (*ΔBT3348*, orange) harboring P-*BT3332* following the introduction of an equal mixture of CS and galactose normalized to measurements from identical strains supplied galactose alone. **c** Relative luminescence from *wild-type Bt* (black) or strains lacking 3 CS-specific

lyases (*ΔBT3324 ΔBT3350 ΩBT4410*, purple) or the CS-sensor (*ΔBT3334*, red) harboring P-*BT3332* following the introduction of a mixture of unsulfated CS disaccharide (diOS) and galactose normalized to measurements from identical strains supplied galactose alone. Values are the mean of 6 biological replicates and error is SEM in color-matched shading. **d** Relative luminescence from *wild-type Bt* (solid lines) or strains lacking the CS-sensor (*ΔBT3334*, dashed lines) harboring P-*BT3332* following the introduction of a mixture of galactose and either hyaluronic acid (HA, green) or heparin (blue) and galactose normalized to measurements from identical strains supplied galactose alone. For **b, d**, values are the mean of 8 biological replicates and error is SEM in color-matched shading. Source data are provided as a Source Data file.

To further optimize luciferase activity, we re-organized the *lux* operon to *luxCDEAB* and exchanged the *Pl* intergenic regions for the *Bt* intergenic regions from *BT1160-1155* (Fig. 2a), which increased luminescence output approximately tenfold over the *Pl* cassette without altering growth kinetics (Fig. 2b and Supplementary Fig. 3a). A *Bt* strain harboring the *Bacteroides*-optimized *lux* cassette exhibited consistently higher relative luminescence than a strain expressing the *Pl* cassette during logarithmic growth (Fig. 2b and Supplementary Fig. 3a) and remained readily detectable even in late stationary phase over 48 h (Fig. 2b). Although luminescence can be detected from a single copy reporter (Supplementary Fig. 3b), the activity reached the baseline by 30 h compared to a multi-copy plasmid that was detectable over 48 h (Fig. 2b). Therefore, we constructed a plasmid containing tandem BamHI and SpeI restriction sites upstream of rpiL*-*luxCDEAB* (p*Bolux*, Fig. 2a) for efficient cloning of promoters. A *Bt* strain harboring p*Bolux* exhibited similar sustained luminescence over 18 h in galactose (Fig. 2c), glucose, fructose, arabinose, or xylose as sole carbon sources (Supplementary Fig. 3c) and introduction of the *Bt rpoD* promoter into the BamHI and SpeI sites increased activity tenfold independently of carbon source (Fig. 2c and Supplementary Fig. 3c). Introducing p*Bolux* into *Bo* produced similar results across all conditions and a plasmid preceded by the region upstream of the *Bo rpoD* gene (*BACOVA_00615*) increased luminescence approximately tenfold (Supplementary Fig. 3d, e). These data indicate that p*Bolux* can reflect promoter-dependent transcription in multiple *Bacteroides* species during anaerobic culture with minimal cost to bacterial growth.

### Engineering a glycan-responsive PUL reporter
To determine if p*Bolux* can accurately indicate changes in PUL transcription in response to target glycans, we introduced the 300 bp region immediately upstream of the *susC* gene from the CS PUL (P-*BT3332*)[16]. We measured luminescence in a *wild-type Bt* strain harboring P-*BT3332* or promoter-less p*Bolux* for 18 h following the introduction of CS or galactose as the sole carbon source (Supplementary Fig. 4a). After only 2 h, the relative luminescence from a strain harboring P-*BT3332* was 46-fold greater and reached a peak activity of 65-fold by 7 h following the introduction of CS compared to cultures supplied galactose alone (Fig. 3a). Conversely, a strain harboring p*Bolux* exhibited identical luminescence in CS relative to galactose until 6 h where it slowly increased sevenfold over 18 h (Fig. 3a). Although introducing multiple copies of the *BT3332* promoter region

reduced endogenous *susC* expression (Supplementary Fig. 4b), which is required for growth on CS (Fig. 1c), this imposed no detectable growth defects on CS (Supplementary Fig. 4c), indicating that increased luminescence carries minimal fitness cost.

To examine whether genes encoded within the CS PUL (*BT3328-BT3334* and *BT4410-BT4411*) are required for the observed P-*BT3332*-specific luminescence increases in CS, we compared reporter activity in mutant *Bt* strains deficient for CS transport, depolymerization, and detection activities necessary for growth on CS (Fig. 1c) following the introduction of a mixture of 0.2% CS and 0.2% galactose (Supplementary Fig. 4d) (galactose was added as a supplemental carbon source to support growth and luciferase activity) or galactose alone (Supplementary Fig. 4e). We determined that a mutant lacking the CS-sensor, *BT3334*, exhibited no increased luminescence over 18 h following the introduction of CS (Fig. 3b). The reduced bioluminescence exhibited by this strain was not the consequence of disabling growth on CS because a mutant lacking a glucuronyl hydrolase, *BT3348* (*gh*), is unable to utilize CS (Fig. 1c), but exhibits sensor-dependent PUL transcription[16] and increased reporter activity similar to *wild-type Bt* (Fig. 3b). These data suggest that disabling genes necessary to generate the PUL-sensor ligand, the 6- or unsulfated CS-derived disaccharide (diOS)[16], in the periplasm prevent sensor-dependent reporter activity increases. Consistent with this notion, a mutant lacking the CS PUL-encoded *susC* transporter, *BT3332*, exhibited a dramatic reduction in reporter activity with a peak activity of 3.1-fold increase at 10 h putatively by reducing CS importation from the extracellular environment (Fig. 3b). Furthermore, disabling 3 CS-specific polysaccharide lyases necessary for the conversion of CS polymers into diOS (BT3324, BT3350, and BT4410) also dramatically reduced reporter activity in response to CS relative to *wild-type Bt*, peaking at 3.3-fold after 1 h following the introduction of CS (Fig. 3b)[16,20]. However, supplying diOS increased luminescence 39.6-fold by 16 h in the lyase-deficient but not the sensor-deficient mutant (Fig. 3c). Collectively, these data indicate that a strain harboring P-*BT3332* can indicate the presence of CS in a manner dependent on PUL-encoded glycan importation, depolymerization, and detection activities.

To determine whether increased activity from a strain containing P-*BT3332* were limited to CS-PUL substrates, we examined luminescence following the addition of a structurally similar but compositionally distinct glycosaminoglycan, heparin (Supplementary Fig. 4f),

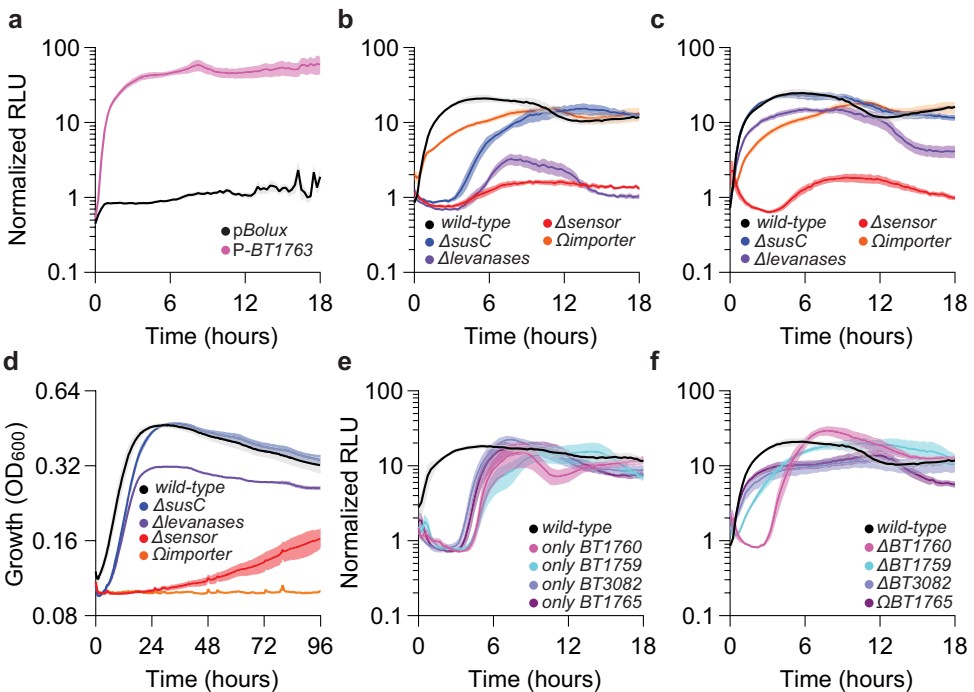

**Fig. 4 | A levan-responsive reporter reveals multiple levanases coordinate fructan utilization in Bt. a** Relative luminescence from *wild-type Bt* harboring p*Bolux* (black) or a plasmid including the promoter region preceding the levan-inducible *susC* gene (P-*BT1763*, pink) following the introduction of levan as the sole carbon source and normalized by the relative luminescence of identical cultures supplied galactose. Values are the mean of 12 biological replicates and error is SEM in color-matched shading. **b, c** Relative luminescence from wild-type *Bt* (black) or strains lacking the levan-inducible *susC* (*ΔBT1763*, blue), 4 levan-specific hydrolases (*ΔBT1760-1759 ΔBT3082 ΩBT1765*, purple), fructan sensor (*ΔBT1754*, red), or a putative inner-membrane fructose transporter (*ΩBT1758*, orange) harboring P-*BT1763* were measured following the introduction of an equal mixture of galactose and (**b**) levan or (**c**) fructose and normalized by the relative luminescence of identical cultures supplied galactose alone. **d** Growth of *wild-type Bt* (black) or strains lacking the levan-inducible *susC* (*ΔBT1763*, blue), 4 levan-specific hydrolases

(*ΔBT1760-1759 ΔBT3082 ΩBT1765*, purple), fructan sensor (*ΔBT1754*, red), or a putative inner-membrane fructose transporter (*ΩBT1758*, orange) were measured during anaerobic culture in minimal media containing fructose as a sole carbon source. **e** Relative luminescence of *wild-type Bt* or strains lacking all other levanases except *BT1760* (*ΔBT1759 ΔBT3082 ΩBT1765*, pink), *BT1759* (*ΔBT1760 ΔBT3082 ΩBT1765*, teal), *BT3082* (*ΔBT1760-59 ΩBT1765*, lavender), or *BT1765* (*ΔBT1760-59 ΔBT3082*, purple) harboring the levan-responsive reporter following the intro-duction of a mixture of levan and galactose normalized with measurements from identical cultures supplied galactose alone. **f** Relative luminescence of *wild-type Bt* or strains lacking *BT1760* (pink), *BT1759* (teal), *BT3082* (lavender), or *BT1765* (pur-ple) harboring P-*BT1763* following the introduction of a mixture of levan and galactose normalized with measurements from identical cultures supplied galac-tose alone. For **b–f**, values are the mean of 8 biological replicates and error is SEM in color-matched shading. Source data are provided as a Source Data file.

which is consumed by *Bt* independently of the CS-PUL (Supplementary Fig. 4g)[34]. As expected, a strain harboring P-*BT3332* exhibited a dra-matically lower activity following the addition of heparin compared to the addition of CS (Fig. 3d), peaking at 2.6-fold after an hour. Con-versely, the addition of hyaluronic acid (HA), which also activates sensor-dependent CS-PUL gene expression required for HA utilization (Supplementary Fig. 4h)[16], increased reporter activity similarly to CS in *wild-type Bt* but not the sensor-deficient mutant (Fig. 3d and Supple-mentary Fig. 4i). Thus, the *Bacteroides*-optimized *lux* reporter can specifically indicate the presence of target glycans by reflecting PUL sensor-dependent responses to glycan-derived ligands.

**A fructan-responsive reporter reveals insights into *Bt* levan utilization**

*Bt* consumption of the polyfructan, levan, requires a distinct PUL encoded by *BT1754-BT1765*[19] and includes an unlinked but co-regulated exo-levanase, *BT3082*[19]. A *Bt* strain harboring p*Bolux* containing the 300 bp region preceding the corresponding *susC* gene (P-*BT1763*) exhibits 35-fold increased activity 5.5 h after the introduction of levan compared to galactose (Fig. 4a and Supplementary Fig. 5a). Levan-dependent reporter activity requires the *BT1763* promoter region because a strain harboring the promoter-less plasmid, p*Bolux*, exhibits no change after 18 h following the introduction of levan (Fig. 4a and Supplementary Fig. 5a). Strains harboring P-*BT1763* or p*Bolux* exhibited identical growth kinetics in either galactose (Sup-plementary Fig. 5b) or levan (Supplementary Fig. 5c) as sole carbon

sources, even though the introduction of multiple copies of the *BT1763* promoter region reduced endogenous *susC* expression in both con-ditions (Supplementary Fig. 5d). These results indicate that differences in reporter activity are not attributable to plasmid-dependent growth effects or altered expression of PUL genes required for levan utiliza-tion. Reporter activation requires levan-derived fructose detection by the PUL sensor protein, BT1754, because a mutant lacking *BT1754* displays only 1.4- or 1.7-fold increased reporter activity following the introduction of either levan (Fig. 4b) or fructose (Fig. 4c), respectively[19]. Moreover, a strain lacking the corresponding *susC* gene exhibited a 3 h delay before reporter activity increased following the addition of levan (Fig. 4b) and 60 h lag before detectable growth on levan as the sole carbon source (Fig. 1e). Conversely, identical strains exhibit reporter activity (Fig. 4c) and growth kinetics (Fig. 4d) resem-bling *wild-type Bt* when supplied fructose, confirming that the fructan-specific SusC is dispensable for transport of monomeric fructose and internalizes levan-derived fructo-oligosaccharides[19,35]. Consistent with this notion, we determined that inactivation of 4 levanases (*BT1760*, *BT1759*, *BT1765* and *BT3082*) reduced levan-dependent reporter activity to levels similar to the *BT1754*-deficient strain (Fig. 4b) and abolished utilization of levan as the sole carbon source (Fig. 1e). Strains harboring only a single levanase, generated by disabling various combinations of all other 3 levanases, exhibited reporter activity increases in levan after a 4 h delay (Fig. 4e) and grow on levan as the sole carbon source (Supplementary Fig. 6a) albeit at dramatically reduced rates and maximum cell densities compared to *wild-type Bt*.

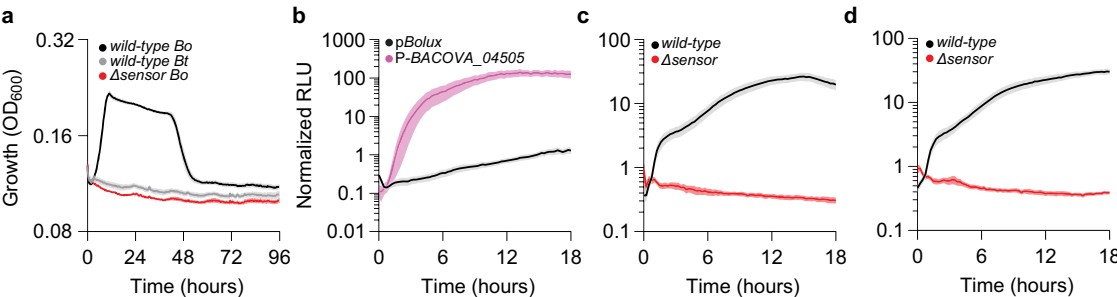

**Fig. 5 | Species-specific responses enable PUL reporters to distinguish between compositionally identical yet structurally distinct glycans. a** Growth of *wild-type Bt* (gray) and *Bo* (black) or a strain lacking the *Bo* inulin sensor (*ΔBACOVA_04496*, red) were measured during anaerobic culture in minimal media containing inulin as a sole carbon source. **b** Relative luminescence from *wild-type Bo* harboring p*Bolux* (black) or a plasmid including the promoter region preceding the inulin-inducible *susC* gene (P-*BACOVA_04505*, pink) were measured following the introduction of inulin as the sole carbon source and normalized by the relative luminescence from identical cultures supplied galactose. Values are the mean of 12 biological replicates and error is SEM in color-matched shading. Relative luminescence from *wild-type Bo* harboring P-*BACOVA_04505* (black) or an isogenic strain lacking the *Bo* inulin sensor (*ΔBACOVA_04496*, red) were measured following the introduction of an equal mixture of galactose and (**c**) inulin or (**d**) fructose and normalized by the relative luminescence of identical cultures supplied galactose alone. For panels **a, c, d**, values are the mean of 8 biological replicates and error is SEM in color-matched shading. Source data are provided as a Source Data file.

Unexpectedly, all strains lacking the exo-levanase, *BT1765*, exhibited reduced maximum cell densities during growth in levan (Supplementary Fig. 6a, b) or fructose (Supplementary Fig. 6c, d) as the sole carbon source, and this was specific because *BT1765*-deficient strains grew similarly to *wild-type Bt* in galactose (Supplementary Fig. 6e, f). Finally, reduced levan-dependent reporter activity in any mutant is not due to diminished growth because a mutant lacking the inner-membrane fructose importer, *BT1758*, cannot grow on levan (Fig. 1e) or fructose (Fig. 4d) as sole carbon sources but exhibit reporter activity resembling *wild-type Bt* in response to either carbohydrate (Fig. 4b, c, respectively). Collectively, these data demonstrate that P-*BT1763* containing strains indicate the presence of levan-derived fructose in a manner requiring PUL-encoded transport, depolymerization, and detection proteins independently of its utilization.

Our results reveal that *Bt* employs multiple levanases to consume levan and demonstrate that three distinct exo-levanases can function independently of the endo-levanase, BT1760, contrasting previous conflicting reports asserting this gene was either essential[19] or dispensable[35] for growth on levan. To examine the role of BT1760 in levan utilization, we independently constructed a *BT1760*-deficient strain, which exhibited reduced growth on levan as the sole carbon source (Supplementary Fig. 6f) and delayed reporter activation in response to levan compared to *wild-type Bt* (Fig. 4f). This mutant exhibited growth on levan resembling *wild-type Bt* when complemented in trans (Supplementary Fig. 6g) indicating that this *BT1760* mutation did not disrupt expression of downstream genes required for levan utilization (Fig. 1e). Furthermore, a mutant encoding only *BT1760* and lacking all known exo-levanases (*ΔBT1759, ΔBT3082 ΩBT1765*), exhibited the greatest delay in reporter activity increases (Fig. 4e) and achieved the lowest maximum growth on levan as the sole carbon source (Supplementary Fig. 6a) compared to any other combination of levanase mutations. Collectively, these data demonstrate that exo-levanases function independently of BT1760 to liberate fructose from levan highlighting that *Bt* levan utilization is not completely understood. Finally, these results illustrate how the p*Bolux* reporter plasmid can be implemented to genetically dissect the contributions of individual PUL-encoded activities to target glycan consumption.

## Species-specific reporter responses distinguish between compositionally identical glycans

*Bacteroides* species can differentially consume structurally distinct glycans comprised of identical monosaccharide components[3,19]. This is exemplified by the fructan PUL, which is conserved across several species and confers *Bt* the ability to consume the β(2,6)-linked polyfructan, levan (Fig. 1e), but not the β(2,1)-linked polyfructan, inulin (Fig. 5a)[11,19]. Conversely, a similar PUL in *Bo* (*BACOVA_04496-BACOVA_04507*) confers inulin (Fig. 5a) but not levan utilization (Supplementary Fig. 7a)[3,11,19]. *Bo* inulin utilization requires the PUL-sensor protein, BACOVA_04496, because a mutant lacking the corresponding gene is unable to grow on inulin as a sole carbon source (Supplementary Fig. 7a) and exhibits reduced growth rates on fructose (Supplementary Fig. 7b) but not galactose (Supplementary Fig. 7c), suggesting that the *Bo* fructan PUL sensor detects monomeric fructose similarly to the *Bt* fructan PUL sensor. To examine how similar PULs encoded by closely related species can confer access to compositionally identical yet structurally distinct glycans, we generated a *Bo* strain harboring a reporter plasmid containing the 300 bp region upstream of the corresponding *susC* gene (P-*BACOVA_04505*). This strain exhibits dramatically increased activity following the addition of inulin compared to galactose, in contrast to a promoter-less control plasmid (Fig. 5b and Supplementary Fig. 7d). Increased reporter activity in *Bo* strains harboring P-*BACOVA_04505* requires the fructan PUL sensor because no change in reporter activity was observed when the sensor mutant was supplied inulin (Fig. 5c) or fructose (Fig. 5d). Collectively, these results indicate that the *Bo* fructan PUL sensor responds to monomeric fructose in a manner similar to the *Bt* sensor although each PUL confers access to structurally distinct glycans. Thus, identical PUL sensor specificities are employed to mediate utilization of distinct fructans suggesting that other PUL-encoded products facilitate species-specific growth on inulin and levan.

Interestingly, the addition of levan elicits increased reporter activity in *wild-type Bo* harboring P-*BACOVA_04505* compared to a strain lacking the corresponding PUL sensor (Supplementary Fig. 7e) although *Bo* is unable to grow on levan as a sole carbon source (Supplementary Fig. 7a). Similarly, *wild-type Bt* harboring P-*BT1763* exhibited detectable reporter activity when supplied with inulin compared to a sensor-deficient strain (Supplementary Fig. 7f), suggesting that *Bo* and *Bt* can derive fructose from levan and inulin, respectively, in quantities insufficient to support growth. We hypothesize that these responses reflect slow, non-specific, and extracellular liberation of fructose from non-target fructans because a *Bt* strain lacking the fructan PUL-encoded *susC* gene, *BT1763*, exhibits identical reporter activity relative to the *wild-type* strain following the addition of fructose (Fig. 4c) or inulin (Supplementary Fig. 7f). Although inulin-dependent reporter activity was substantially reduced in a strain lacking 4 levanases, luminescence levels were greater than a *sensor*-deficient mutant suggesting *Bt* possesses unknown inulin-degrading activities that can likely liberate fructose monomers slowly and non-specifically compared to *Bo* (Supplementary Fig. 7f). Collectively, these results demonstrate how PUL reporters exhibit highly sensitive

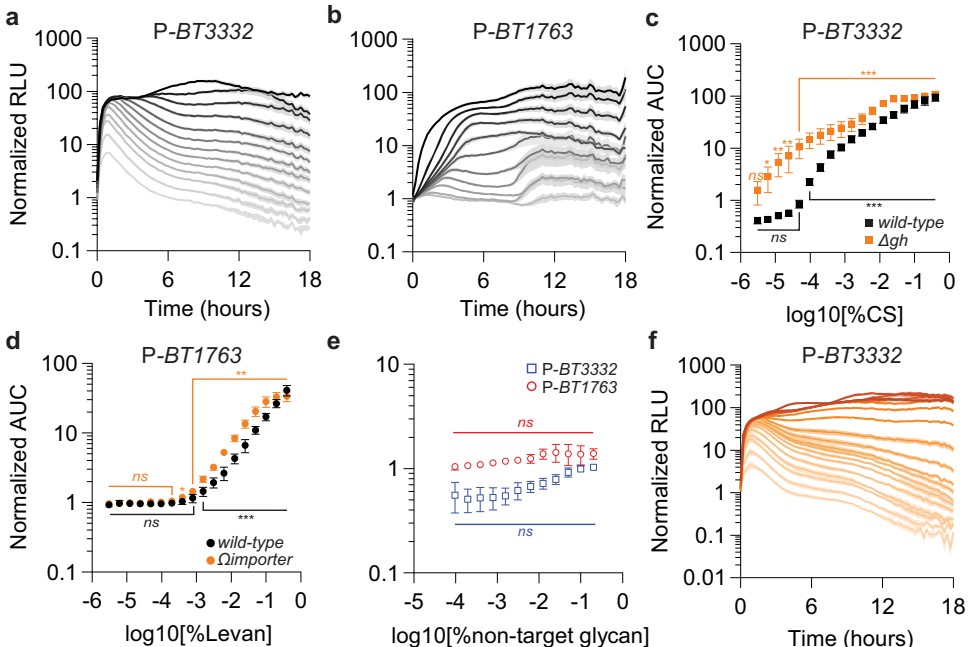

**Fig. 6 | PUL reporters reflect dose-dependent transcription. a** Relative luminescence from a *wild-type Bt* strain harboring P-*BT3332* following the introduction of twofold serial dilutions of 0.4% CS containing galactose to a total carbohydrate content of 0.5% and normalized to identical cultures supplied galactose alone. **b** Relative luminescence from a *wild-type Bt* strain harboring P-*BT1763* following the introduction of twofold serial dilutions of 0.4% levan containing galactose to a total carbohydrate content of 0.5% and normalized to identical cultures supplied galactose alone. For **a** and **b**, values are the mean of 12 biological replicates and error is SEM in color-matched shading. **c** The AUC of response curves measured from *wild-type Bt* (black, see **a**) or a mutant lacking the glucuronyl hydrolase (*Δgh*, orange, see **f**) harboring P-*BT3332* supplied twofold dilutions of CS to each strain and normalized by identical cultures supplied galactose alone. **d** The AUC of response curves measured from *wild-type Bt* (black, see **b**) or a mutant lacking a putative inner-membrane transporter (*Ωimporter*, orange; see Supplementary Fig. 9f) harboring P-*BT1763* supplied twofold dilutions of levan to each strain and normalized by identical cultures supplied galactose alone. For **c, d**, values are the mean of 12 biological replicates, error bars are standard deviation, and *P*-values were computed by two-way ANOVA with Dunnett correction and *** indicates values <0.001, ** <0.01, * <0.05, and ns > 0.05. **e** The AUC of response curves measured from *wild-type Bt* strains harboring either P-*BT3332* (open blue squares) supplied mixtures containing twofold serial dilutions of levan or P-*BT1763* (open red circles) supplied twofold serial dilutions of CS with galactose to a total carbohydrate content of 0.5% and normalized by identical cultures supplied galactose alone. Values are the mean of 6 biological replicates, error bars represent standard deviation, and *P*-values were computed using two-way ANOVA with Tukey's honest significance test and ns indicates values >0.05. **f** Relative luminescence from a *gh*-deficient *Bt* strain harboring P-*BT3332* following the introduction of twofold serial dilutions of 0.4% CS containing galactose to a final carbohydrate content of 0.5% and normalized to identical cultures supplied galactose alone. Values are the mean of 12 biological replicates and error is SEM in color-matched shading. Source data are provided as a Source Data file.

responses in distinct gut *Bacteroides* species and can detect the presence of compositionally identical but structurally distinct glycans independently of their utilization.

## PUL reporters exhibit linear dose-dependent responses

To determine if dose-dependent PUL transcriptional responses elicited by target glycans (Fig. 1g and Supplementary Fig. 2a) are reflected by the corresponding PUL promoter-dependent reporter transcription and activity, we supplied decreasing amounts of either CS or levan to *Bt* strains harboring either the CS/HA- or fructan-responsive reporter plasmids described above. The introduction of either the promoter region preceding *BT3332* or *BT1763* into p*Bolux* conferred kinetic dose-dependent changes in both *luxC* expression and corresponding luminescence following the respective addition of CS (Supplementary Fig. 8a) or levan (Supplementary Fig. 8b). We computed the area under each curve (AUC) for both conditions over 18 h to aggregate the collective output of *lux* gene transcription, translation, and biochemical activities over time, revealing exponential responses from strains harboring P-*BT3332* or P-*BT1763* supplied dilutions of CS (Fig. 6a) or levan (Fig. 6b), respectively, and these responses were significant across more than 3 orders of magnitude. For example, a *wild-type Bt* strain containing P-*BT3332* exhibited increased reporter activity when supplied between 0.0001% (1 μg/mL) and 0.4% (4 mg/mL) CS relative to identical cultures supplied only galactose (Fig. 6a). Similarly, a *wild-type Bt* strain containing P-*BT1763* displayed concentration-dependent

reporter activity increases when supplied between 0.0016% (16 μg/mL) and 0.4% levan (Fig. 6b). The corresponding AUC from each condition revealed that the CS- and levan-reporter responses increase exponentially within these ranges (Fig. 6c, d) and the logarithm of corresponding AUC values follows a linear regression model with an $R^2$ of 0.95 and 0.97, respectively, when supplied their target ligands (Supplementary Fig. 9a, b). Reporter strains respond specifically to their corresponding target glycan because neither the CS- or levan-responsive strains exhibited significantly increased luminescence when supplied any concentration of levan or CS, respectively (Fig. 6e). These data suggest that the exponential, concentration-dependent reporter activity increases exhibited by engineered *Bt* strains in response to their target glycans could be used to estimate target glycan abundance in an unknown sample concentration. However, bacterial species including *Bt* exhibit hierarchical glycan utilization, resulting in prioritized expression of genes necessary for the consumption of more- over less-preferred substrates[17,22,24,36,37], potentially hindering target glycan detection in PUL-reporter strains. However, the CS-reporter strain produced similar dose-dependent responses and exhibited an identical linear range of detection in the presence or absence of levan (Supplementary Fig. 9c, d). Interestingly, the levan-responsive strain exhibited greater reporter activity responses to levan in the presence of 0.2% CS (Supplementary Fig. 9c) expanding the lower range of responsiveness down to 0.0004% (4 μg/mL) levan (Supplementary Fig. 9d) and suggesting that the presence of CS may

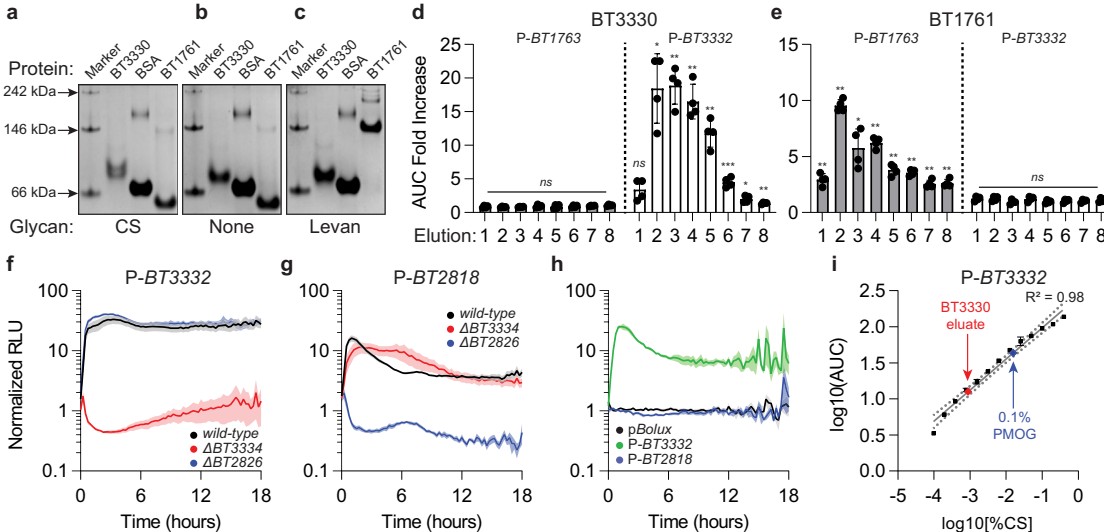

**Fig. 7 | PUL reporters facilitate SGBP-mediated target glycan isolation.** Purified BT3330, BSA, or BT1761 was analyzed by affinity-PAGE in the presence of (**a**) 0.1% CS, (**b**) no glycan or (**c**) 0.1% levan. Images are representative of 3 independent experiments. The fold difference in AUC from *wild-type* Bt strains harboring either P-*BT1763* (gray bars) or P-*BT3332* (open bars) supplied fractions eluted from immobilized (**d**) BT3330 or (**e**) BT1761 in combination with 0.2% galactose and normalized against cultures supplied galactose alone. Normalized AUC values were divided by responses from identical strains supplied equivalent fractions eluted from columns containing control lysates prepared from cells harboring an empty vector (pT7-7; see Supplementary Fig. 10a, b). Values are the mean of 4 biological replicates, error bars are SEM, and *P*-values were computed using a two-tailed student's t-test and *** indicate values <0.001, ** <0.01, * <0.05, and ns > 0.05. Relative luminesence from *wild-type Bt* or strains lacking *BT3334* (red) or *BT2826* (blue) harboring either (**f**) P-*BT3332* or (**g**) P-*BT2818* supplied an equal mixture of PMOG and galactose and normalized to identical cultures supplied galactose alone. Values are the mean of 8 biological replicates and error bars are SEM in color-matched shading. **h** Relative luminesecence from *wild-type Bt* strains harboring p*Bolux* (black), P-*BT3332* (green), or P-*BT2818* (blue) supplied 0.4% galactose and a 1:10 dilution of pooled, concentrated material co-purifying with BT3330 pre-incubated with PMOG and normalized to identical cultures supplied galactose alone. Values are the mean of 4 biological replicates and error bars are SEM in color-matched shading. **i** The AUC from a *wild-type Bt* strain harboring P-*BT3332* supplied 0.4% galactose containing either 0.1% PMOG (blue) or a 1:10 dilution of pooled, concentrated material co-purifying with BT3330 (red) pre-incubated with PMOG with and normalized to identical cultures supplied galactose alone. Measurements were collected alongside an identical strain supplied twofold serial dilutions of CS with galactose to a total carbohydrate content of 0.5% and normalized to identical cultures supplied galactose alone. Linear regression models (gray line) were computed in Prism, and sample concentrations were estimated with the derived equations. Values are the mean of 2 independent measurements, error bars are standard deviation, and the dashed gray lines are the 95% confidence interval. Source data are provided as a Source Data file.

enhance luciferase activity. Consistent with this notion, the addition of 0.2% CS significantly increased the levan-responsive reporter activity when supplied with 0.2% levan (Supplementary Fig. 9e) but did not significantly change fructan-reporter activity in the absence of levan (Fig. 6e). Taken together, these data suggest that PUL reporters can indicate the presence of target substrates in glycan mixtures.

Sensor-dependent PUL transcription by target glycans is typically resolved by negative feedback mechanisms whereby PUL-encoded activities remove the glycan-derived, PUL-sensor ligand from the periplasm, thereby reducing PUL sensor activation (Fig. 1a). For example, the CS-responsive PUL relies on the glucuronyl hydrolase (GH), BT3348, to hydrolyze diOS into its monosaccharide components N-acetyl galactosamine and 5-keto 4-deoxyuronate, thereby removing the sensor ligand and reducing transcription activation[16]. Therefore, a mutant lacking *gh* stimulates increased CS-sensor stimulation and exhibits increased PUL transcription relative to *wild-type Bt*[16]. Accordingly, a *gh*-deficient strain harboring P-*BT3332* exhibits detectable responses between 0.000006% (60 ng/mL) and 0.4% CS (Fig. 6f) but the linearity of these responses reached a maximum at 0.0125% (125 μg/mL) (Fig. 6c and Supplementary Fig. 9a). Similarly, a strain lacking the inner-membrane fructose importer, *BT1758*, harboring P-*BT1763* exhibited increased luminescence at levan concentrations as low as 0.0004% (Supplementary Fig. 9f) but achieved its maximum at 0.2% levan (Fig. 6d) limiting the upper range of its responsiveness (Supplementary Fig. 9b). Thus, PUL-reporter activity can be genetically tuned to extend target glycan sensitivity by preventing PUL-sensor deactivation, which horizontally shifts the glycan detection range. Collectively, our data suggest PUL reporters offer specific and highly sensitive glycan detection reagents that could be implemented to

estimate glycan abundances across detection ranges spanning greater than 1000-fold.

## PUL reporters facilitate target glycan isolation using SGBPs

PULs often encode SGBPs that facilitate glycan sequestration along the outer membrane (Fig. 1a)[19,20,38]. SGBPs specifically bind their cognate glycan ligands and can discriminate between compositionally similar, structurally distinct glycans[19,20,38]. For example, BT1761, an SGBP encoded in the *Bt* levan utilization PUL, binds to levan but not inulin[19] and BT3330, an SGBP encoded in the *Bt* CS utilization PUL differentially binds CS polymers greater than 20 disaccharide units in size[20]. Consistent with these findings, recombinant BT3330 protein exhibited a reduced migration following affinity-PAGE in the presence of high molecular weight CS polymers (Fig. 7a) but not in their absence (Fig. 7b). Conversely, BT1761 protein exhibited a decreased relative migration following affinity-PAGE in the presence of levan (Fig. 7c) but not in its absence (Fig. 7b). Furthermore, the altered migration of both proteins was specific to the presence of target glycans because BT3330 migration was similar in the presence of levan (Fig. 7c) relative to gels lacking glycan (Fig. 7b) and BT1761 migration in CS-containing gels (Fig. 7a) was similar in the absence of glycan (Fig. 7b). These results indicate that each PUL-encoded SGBP can distinguish between target and non-target glycans in vitro and putatively facilitate target glycan isolation from a heterogenous mixture. We reasoned that because PUL-reporter strains exhibit sensitive (Fig. 6c, d) and specific (Fig. 6e) detection of target glycans even in the presence of non-target molecules (Supplementary Fig. 9c, d), they could serve as glycan detection reagents to identify fractions enriched for glycans of interest using immobilized, recombinant SGBPs. Thus, we combined cell lysates

prepared from *Escherichia coli* strains engineered to over-express either N-terminally his-tagged BT1761 or BT3330 with an equal mixture of levan and CS. After incubation, SGBPs were captured using Ni-agarose, washed extensively, and subsequently eluted with free histidine so that each fraction could be supplied directly to *Bt* cultures harboring either P-*BT3332* or P-*BT1763*. Elutions 2 through 8 from columns containing BT3330 elicited significantly increased CS-responsive reporter activity compared to control reactions incubated with lysates prepared from *E. coli* strains harboring an empty vector (Fig. 7d and Supplementary Fig. 10a). Conversely, identical fractions supplied to the fructan-responsive reporter strain resulted in activity that was indistinguishable from those eluted from vector control containing lysates (Fig. 7e and Supplementary Fig. 10b) indicating that BT3330-containing fractions (Supplementary Fig. 10c) were enriched for CS over levan. Fractions co-eluting with BT1761 (Supplementary Fig. 10c) elicited increased activity from the levan- (Fig. 7e and Supplementary Fig. 10b) but not the CS-responsive reporter strain relative to elutions from a vector control (Fig. 7e and Supplementary Fig. 10a), indicating BT1761 enriched levan over CS.

To measure SGBP-mediated target glycan enrichment, we concentrated pooled fractions co-eluting with either BT3330 or BT1761 that elicited significantly increased reporter activity compared to control fractions (Fig. 7d, e). Supplying the concentrated material that co-purified with BT3330 to *Bt* reporter strains elicited increased activity from a *wild-type Bt* strain harboring P-*BT3332* (Supplementary Fig. 10d) but not P-*BT1763* (Supplementary Fig. 10e). Conversely, material co-purifying with BT1761 stimulated levan- (Supplementary Fig. 10e) but not CS-responsive reporter activity (Supplementary Fig. 10d). To determine whether extending the lower limits of detection could indicate the enrichment for each target glycan, we supplied the concentrated eluates to hypersensitive strains defective for sensor deactivation (Supplementary Fig. 6c, d). The concentrated eluates from BT1761 stimulated fourfold increased reporter activity in the *gh*-deficient strain harboring P-*BT3332* (Supplementary Fig. 10f), suggesting that some CS co-purified with the levan-specific SGBP, however, this increase was not significant. Because the linear range of detection extends to 60 ng/mL CS (Supplementary Fig. 9a), we estimate that BT1761 was able to enrich levan greater than 1000-fold over CS. Similarly, the concentrated BT3330 eluate did not significantly increase reporter activity in the fructose importer-deficient strain harboring P-*BT1763* (Supplementary Fig. 10g) indicating that BT3330 enriched CS at least 50-fold over levan.

By comparing each measurement to a standard curve of twofold CS or levan dilutions, we calculated that the BT3330 eluate contained 228.5 μg/mL CS and the BT1761 eluate contained 271.6 μg/mL levan using a simple linear regression of log10 reduced AUC values (Supplementary Fig. 10h, i, respectively). To verify these results with reporter-independent methods, we measured total fructan or glycosaminoglycan content from identical samples using colorimetric assays, which indicated that concentrated eluates co-purifying with BT3330 containing 168.2 μg/mL CS (Supplementary Fig. 10j) but no detectable levan and material co-purifying with BT1761 contained 227.3 μg/mL levan (Supplementary Fig. 10k) but no detectable CS. Collectively, these data demonstrate that PUL reporters are powerful detection tools that can facilitate target glycan isolation from mixtures and measurement beyond the limits of commercially available solutions.

To determine if the tools and methods described in this work could be applied to a biologically derived glycan mixture, we examined responses from strains harboring P-*BT3332* following the addition of porcine mucosal O-glycans (PMOG), which are derived from the gastric mucosa, contain 13.1% glycosaminoglycans (Supplementary Fig. 11a) and increase *BT3332* transcript levels when supplied to *Bt*[9]. As expected, a *wild-type Bt* strain containing P-*BT3332* exhibits 33.4-fold

increased luminescence by 3.25 h following the introduction of 0.2% PMOG, and this requires the corresponding sensor gene, *BT3334* (Fig. 7f). Because PMOG elicits increased transcription of additional *Bt* PULs including one spanning *BT2818-BT2825*[9], whose expression is controlled by the adjacent sensor gene, *BT2826*[39], we engineered a reporter plasmid that includes the 300 bp promoter region preceding the *susC*-like gene, *BT2818* (P-*BT2818*). A *wild-type Bt* strain harboring P-*BT2818* exhibited 16.2-fold increased luminescence peaking at 1 h following the introduction of PMOG and this response required *BT2826* (Fig. 7g). The *BT2818-BT2826* PUL targets an unknown glycan in PMOG distinct from CS because: (1) loss of the CS-sensor, BT3334, does not dramatically alter P-*BT2818* responses to PMOG (Fig. 7g); (2) addition of CS diOS did not increase luminescence from a strain harboring P-*BT2818* (Supplementary Fig. 11b); (3) strains harboring mutations in *BT2818-BT2826* exhibit no competitive defects in CS as a sole carbon source;[40] and (4) *BT2826*-deficient and *wild-type* strains harboring P-*BT3332* exhibit identical increases in luminescence following the addition of PMOG (Fig. 7f). Thus, p*Bolux*-derived reporters can be engineered to detect unknown target molecules present in biologically derived heterogeneous glycan mixtures.

To determine if we could employ SGBPs to isolate target glycans from this mixture, we examined responses from strains harboring P-*BT3332* supplied elution fractions collected from a similar purification procedure as described above with lysates containing recombinant BT3330 protein pre-incubated with 0.1% PMOG. Accordingly, elutions 1-7 elicited significant increases in luminescence compared to control reactions (Supplementary Fig. 11c) indicating that BT3330 retained target molecules from PMOG. BT3330 enriched target glycans away from non-targets because pooled, proteolyzed, and concentrated elutions increased luminescence from strains harboring P-*BT3332* but not P-*BT2818* (Fig. 7h), whereas the input PMOG mixture increased both (Fig. 7f and g, respectively). Finally, comparing the AUC from strains harboring P-*BT3332* supplied the BT3330 eluate to those supplied a standard curve of CS, indicated that we recovered 196.7 μg/mL CS/HA from PMOG (Fig. 7i), which was similar to the 121.4 μg/mL estimated by a colorimetric assay (Supplementary Fig. 11a). Collectively, these data demonstrate that SGBPs can enrich target molecules present in undefined heterogeneous biologically derived glycan mixtures and PUL reporters can sensitively and specifically track target glycan retention, determine enrichment, and quantify the recovered products.

## Discussion

This work describes a reporter system that kinetically monitors transcription during anaerobic growth in prominent human gut commensal bacteria. Here, we establish that introducing p*Bolux* plasmids containing PUL promoters generates glycan-sensitive bacterial strains that require PUL-encoded transport, depolymerization, and detection activities (Figs. 3–5). We demonstrate that *Bacteroides* PUL sensors elicit concentration-dependent transcriptional responses (Fig. 1g and Supplementary Fig. 2a) that are recapitulated by corresponding reporter strains (Fig. 6 and Supplementary Figs. S8 and 9) and can indicate the presence of target molecules at concentrations below the limits of detection of commercially available solutions (Supplementary Fig. 10). Moreover, we demonstrate that reporter sensitivity can be genetically tuned to respond by glycan degree of polymerization by disabling glycan depolymerization machinery (Figs. 3 and 4) or achieve lower limits of detection disabling PUL-encoded genes necessary for PUL sensor deactivation (Fig. 6f and Supplementary Fig. 9a, b, f). Finally, we demonstrate that reporter responses are readily quantifiable and can estimate unknown target glycan concentrations (Fig. 7i and Supplementary Fig. 10h, i) similar to reporter-independent methods (Supplementary Figs. 10j, k and 11a). Thus, this work establishes a glycomics toolset that employs *Bacteroides* PULs to facilitate simple, rapid, and high-

throughput detection and quantification of gut microbial substrates in heterogeneous mixtures.

Examining bacterial glycan utilization has previously relied on detectable growth on commercially purified glycans or steady-state transcriptional changes following introduction of undefined glycan mixtures[9–11,17–19,21,22,24]. We demonstrate that PUL reporters can indicate bacterial transcriptional responses to target glycans in dramatically smaller volumes (as low as 30 μL) (Figs. 6 and 7) and require substantially less material than similar measurements using ex vivo approaches such as qPCR, transcriptomics, or NanoLuc[9,11,25]. Furthermore, p*Bolux*-derived reporter strains overcome the need for kinetic sampling of bacterial cultures because changes in gene expression can be measured over time into late stationary phase and can indicate target glycans independently of their utilization (Figs. 1c, e, 3b, and 4b). Importantly, because these strains can be cultured *ad infinitum* and do not require harvesting mRNA or cDNA synthesis, they offer a high-throughput, cost-effective tool to measure changes in gene transcription during anaerobic growth. These advantages render p*Bolux* a powerful, efficient, and relatively inexpensive tool to examine *Bacteroides* gene transcription in various conditions including in response to glycans, altered in vitro growth conditions, and possibly in situ during mammalian intestinal colonization.

To demonstrate the advantages of p*Bolux*-derived plasmids, we leveraged three PUL reporters to devise an affinity isolation approach whereby recombinant PUL-encoded SGBPs isolate target glycans from defined, laboratory prepared and undefined biologically derived mixtures (Fig. 7). Because the PUL-reporter strains can specifically and sensitively detect target glycans even in the presence of non-targets (Fig. 7f, g and Supplementary Fig. 9d, e), we could identify fractions containing two distinct SGBPs co-purifying with their corresponding glycan ligands (Fig. 7d, e and Supplementary Fig. 11c). Furthermore, using strains with extended limits of target glycan detection, we could demonstrate that our approach achieved greater than 1000-fold enrichment of target from non-target molecules (Supplementary Fig. 10f, g). We hypothesize that similar approaches could be implemented to isolate unknown PUL targets such as those derived from the mammalian mucosa, which can exert prebiotic effects when administered in aggregate[41], for downstream structural characterization. In total, we have harnessed the dedicated glycan utilization machinery possessed by gut microbes to develop tools that enable simple, efficient, and inexpensive glycan detection, isolation, and quantification that are readily scalable and distributable.

Finally, we demonstrate how glycan-responsive reporters can reflect species-specific glycan preferences to detect structurally distinct yet compositionally identical glycans (Figs. 4 and 5). With some *Bacteroides* species encoding over one hundred different PULs, many with unknown glycan targets, the accessible glycome of intestinal microbes is vast and largely uncharacterized[11,15]. Moreover, some annotated PULs putatively facilitate utilization of unknown glycan substrates that confer species-specific gut colonization advantages[42–44], which could represent attractive therapeutic targets and be identified by implementing analogous approaches to those described here. Because PUL reporters can indicate the presence of target glycans at low abundances (Fig. 6d, e and Supplementary Fig. 9a, b), without serving as growth substrates (Figs. 3b, 4b, c, and Supplementary Fig. 7e, f), and even in the presence of other glycans (Fig. 7e, f and Supplementary Fig. 9c–e), PUL reporters could be used to efficiently screen various mixtures for the presence of target glycans. Thus, constructing p*Bolux* plasmids containing promoters from PULs with unknown targets could be used to examine differential glycan abundances across low-yield biological samples in high-throughput applications, enabling glycan surveillance approaches.

## Methods

### Bacterial culture
All *E. coli* strains were cultured on LB agar (BD) aerobically at 37 °C and inoculated from single colonies into LB media (BD) with agitation at 275 rpm. All *Bacteroides* strains were cultured as previously described[44,45] on brain-heart infusion agar (70138, MilliporeSigma) containing 5% horse blood (Hardy) anaerobically and inoculated from single colonies into TYG incubated under identical conditions. *Bacteroides* strains were sub-cultured at the indicated dilutions from stationary phase growth in TYG into minimal media containing the indicated carbon sources described in the corresponding figure legends. All strains were cultured in the presence of antibiotic selection where appropriate at the following concentrations: 100 μg/mL ampicillin, 2 μg/mL tetracycline, 25 μg/mL erythromycin.

### Growth measurements
*Bacteroides* strains were cultured anaerobically to stationary phase in TYG and diluted 1:200 into pre-reduced minimal media containing glucose (G8250, MilliporeSigma), galactose (G0750, MilliporeSigma), fructose (F0127, MilliporeSigma), arabinose (A3256, MilliporeSigma), xylose (X1500, MilliporeSigma), chondroitin sulfate (C9819, MilliporeSigma), heparin (MilliporeSigma, H4784), hyaluronic acid (MilliporeSigma, 53747), levan (P-LEVAN, Megazyme), or inulin (P-INUL, Megazyme) where indicated in pre-reduced 96-well or 384-well clear microplates (Corning). To remove free fructose from inulin, a 1% inulin solution was passed over a ZebaSpin de-salting column (ThermoFisher) prior to the addition of cultures. Kinetic absorbance measurements were taken at 600 nm every 15 min for 96 h using a Tecan Infinite M-plex maintained at 37 °C in a Coy anaerobic chamber with a 2.5% hydrogen atmosphere.

### Strain construction
A plasmid encoding the *Pl lux* cassette under control of the *Bt rpoD* promoter (*BT1311*) was generated by amplifying products from p*mini-Tn5 luxCDABE Tc*[46] using primers W3115 and W3124 and P$_{BT1311}$-rpiL* from strain NS340 with primers W2952 and W2905[17]. The resulting amplicons were assembled with pNBU2-tetQ digested with BamHI-HF (NEB) and SalI-HF (NEB) using the NEBuilder Assembly Master Mix (NEB). The resulting plasmid (pNBU2-*lux-Pl*) was verified by Sanger sequencing before its introduction into *Bt* by di-parental mating to generate strain GT962, and the insert was sub-cloned into pLYL01 using BamHI-HF and SalI-HF to generate p*lux-Pl*.

To generate a *Bt* optimized lux cassette, *luxC* (primers W2952 and W3265), *luxD* (primers W3266 and W3267), *luxE* (primers W3268 and W3269), *luxA* (primers W3270 and W3271) and *luxB* (primers W3272 and W3273) were amplified from GT962 genomic DNA and assembled with pNBU2-tetQ digested with BamHI-HF and SalI-HF using the NEBuilder Assembly Master Mix. The resulting plasmid (pNBU2-*lux-Bt*) was verified by Sanger sequencing before being sub-cloned into pLYL01 using BamHI-HF and SalI-HF to generate p*lux-Bt*. A promoter-less *Bacteroides*-optimized lux cassette was generated by assembling the *lux* cassette from p*lux-Bt*, amplified with primers 1080 and 1011, with pLYL01 digested with BamHI-HF and SalI-HF using the NEBuilder Assembly Master Mix . The resulting plasmid, p*Bolux*, contains tandem BamHI and SpeI restriction sites for cloning and was used for all experiments as a negative control. To construct chondroitin sulfate, levan, and inulin-responsive reporters, the 300 bp regions upstream of *BT3332* (primers 1232 and 1373), *BT1763* (primers 1150 and 1304), *BACOVA_04505* (primers 1943 and 1944), or *BT2818* (primers 1188 and 1351) were amplified from the appropriate genomic DNA and combined with p*Bolux* digested with BamHI-HF and SpeI-HF using the NEBuilder Assembly Master Mix.

Generating loss-of-function mutations in *Bt* were performed as previously described[16] using pEXCHANGE-tdk or pKNOCK-ermGb. A

*Bo* strain lacking *BACOVA_04496* was constructed as previously described[47] using a similar allelic exchange method with pSIE1. Briefly, plasmids targeting each gene (Supplementary Table 3) were introduced by di-parental mating and proper chromosomal integration was verified by PCR. Counter selection against pEXCHANGE-tdk or pSIE1-derived constructs was performed on brain-heart infusion agar containing 5% horse blood media and 200 ng/mL 5-fluoro-2-deoxyuridine (DOT Scientific) or 100 ng/mL anhydrotetracycline (ThermoFisher), respectively. All mutations were verified by Sanger sequencing across the chromosomal region of interest. All strains, primers, and plasmids are listed in Supplementary Tables 1–3, respectively.

## Transcript quantification
mRNA was prepared from 1.0 mL of pelleted *Bt* cell culture treated with RNA protect (Qiagen) using the RNeasy kit (Qiagen) according to the manufacturer's directions. cDNA was subsequently synthesized from 500 ng of isolated RNA using Superscript VILO IV master mix (ThermoFisher) according to the manufacturer's directions. Transcript levels were measured by qPCR using PowerUp SYBR Green PCR Master Mix (Applied Biosystems) and primers 1060 and 1061 (*BT3332*), 1056 and 1057 (*BT1763*), or 2208 and 2209 (*luxC*) were monitored using a QuantStudio 12 K Flex instrument (Applied Biosystems). All mRNA transcripts were normalized as previously described[17] to *16 s* rRNA measured from 1000-fold diluted cDNA using primers 1044 and 1045.

## Measuring bioluminescence during growth in various monosaccharides
*Bt* and *Bo* strains were cultured in TYG containing 2 μg/mL tetracycline overnight before being diluted 200-fold into minimal media containing 0.5% carbon source in a sterile 96-well white, clear-bottomed microplate (Corning 3610). Absorbance at 600 nm and luminescence were measured every 15 min for 48 h using a Tecan Infinite M-plex. RLU were calculated as luminescence values divided by absorbance at 600 nm.

## Measuring bioluminescence in response to glycans
*Bt* and *Bo* strains were cultured in TYG containing 2 μg/mL tetracycline overnight before being diluted 50-fold into minimal media containing 0.5% galactose anaerobically grown to mid-exponential phase at 37 °C (approximately 4 h). Cultured cells were pelleted by centrifugation before the supernatant was removed, and cells were resuspended in 2X minimal media lacking a carbon source. Equal volumes of each cell suspension were transferred to a pre-reduced, white, clear bottom 384-well microplate (Corning 3765) containing equal volume of following polysaccharides: chondroitin sulfate (MilliporeSigma, C9819), heparin (MilliporeSigma, H4784), hyaluronic acid (MilliporeSigma, 53747), levan (Megazyme, P-LEVAN), or inulin (Megazyme, P-INUL) where indicated. Absorbance at 600 nm and luminescence were measured every 15 min for 18 h using a Tecan Infinite M-Plex instrument anaerobically at 37 °C temperature. RLU were calculated as luminescence values divided by absorbance at 600 nm and normalized to identical measurements from each strain supplied only galactose.

## Protein overexpression and purification
*E. coli* BL21 (DE3) cells were transformed with pT7-7 plasmids encoding SGBPs. Fresh transformants were cultured overnight in LB supplemented with 100 μg/mL ampicillin at 37 °C with agitation. The following day, cultures were diluted 50-fold in LB supplemented with 100 μg/mL ampicillin and incubated for approximately 2 h at 37 °C with agitation to mid-exponential phase ($OD_{600}$ ~ 0.45-0.6). Cultures were cooled on ice before isopropyl β-thiogalactopyranoside (MilliporeSigma) was added to a final concentration of 50 μM and incubated at 30 °C with agitation for 4 h. Cells were pelleted by centrifugation at 7197 × *g* for 10 min at 4 °C before the supernatant was decanted and cell pellets were frozen on dry ice and stored at −80 °C. Pellets were

thawed on ice, resuspended in 20 mM Tris/HCl buffer, pH 8.0, containing 100 mM NaCl and lysed in 2.0 mL tubes containing 250 μL volumes of 0.1 mm silica beads placed in a pre-chilled aluminum block using a Bead beater-96 (BioSpec) for 5 cycles of 40 s beating at 2400 rpm and 5 min resting at 4 °C. Lysates were centrifuged at 20,000 × *g* for 10 min at 4 °C and the supernatant was combined with 0.2 mL pre-equilibrated Ni-NTA sepharose and 10 mL of 20 mM Tris/HCl buffer, pH 8.0, containing 100 mM NaCl for 1 h at room temperature with rocking. The slurry was packed into a 2 mL purification column (Pierce) and the liquid phase removed by gravity flow before washing with 3 mL of wash buffer (20 mM Tris/HCl buffer, pH 8.0, containing 100 mM NaCl). SGBPs were eluted from the column with 3 mL of elution buffer (20 mM Tris/HCl buffer, pH 8.0, containing 100 mM NaCl and 25 mM histidine), concentrated and buffer exchanged using centrifugal concentrators (MilliporeSigma) and 6 successive additions of storage buffer (10 mM Tris/HCl buffer containing 10% glycerol). Protein concentrations were estimated by BCA assay (Pierce) and stored at −80 °C.

## Affinity-PAGE
One hundred picomoles of protein was combined with native PAGE loading buffer and electrophoresed in a 10% polyacrylamide matrix containing 10% (w/v) acrylamide in Tris-glycine buffer (pH 8.8) at 20 mA for 3.5 h at room temperature. Gels contained 0.1% CS (MilliporeSigma, C9819) or levan (Megazyme, P-LEVAN) and BSA (Roche, 10711454001) and NativeMark Unstained Protein Standard (ThermoFisher) were used as negative controls. Proteins were visualized by Coomassie Blue staining.

## SGBP target glycan isolation
Five hundred microliters of clarified protein lysate was pre-incubated with 1.0 mL of a mixture containing 0.1% levan and 0.1% CS or 0.1% PMOG overnight at 4 °C with rocking before combination with 200 μL pre-equilibrated Ni-NTA agarose (MilliporeSigma) in 9 mL of 20 mM Tris/HCl buffer, pH 8.0, containing 100 mM NaCl, for 1 h at RT with rocking. The slurry was packed into a 2 mL purification column (Pierce) and the liquid phase removed by gravity flow before washing 4 times with 1 mL of 20 mM Tris/HCl buffer, pH 8.0, containing 100 mM NaCl. SGBPs were eluted from the column in a total of 4 mL elution buffer (20 mM Tris/HCl buffer, pH 8.0, 100 mM NaCl and 25 mM histidine). Glycan content was measured as described above whereby mid-exponential cells were pelleted by centrifugation and re-suspended into 2× minimal media containing 0.4% galactose. Each cell suspension was transferred to a pre-reduced, white, clear bottom 384-well microplate (Corning) containing equal volumes of column fractions. Absorbance and luminescence were measured kinetically under anaerobic conditions as described above. Target glycan-containing elution fractions were combined and treated with 0.1 mg/mL Proteinase K (ThermoFisher) in 10 mM Tris, pH 7.5 containing 20 mM calcium chloride for 2 h at 65 °C. Total glycans were combined with 3 volumes of 200 proof ethanol, incubated overnight with rocking at 4 °C, pelleted by centrifugation, and resuspended in water equivalent to one-tenth the original volume.

## PMOG preparation
PMOG were prepared as previously described[9]. Briefly, porcine gastric mucin (MilliporeSigma, M1778) was suspended in 100 mM Tris (pH 7.4) at 2.5% w/v and autoclaved. After cooling to 55 °C, proteinase K (ThermoFisher) was added to the suspension to a final concentration of 0.1 mg/ml and incubated at 55 °C for 16 h, centrifuged at 7197 × *g* for 30 min at 4 °C to remove insoluble material, and sodium hydroxide was added to the supernatant to a final concentration of 0.1 M. Subsequently, sodium borohydride was added to a final concentration of 1 M and incubated at 65 °C for 18 h before the addition of concentrated hydrochloric acid to achieve a pH of 7.0. The solution was centrifuged

**Article**

at 7197 × *g* for 30 min at 4 °C, passed through a 0.22 μm filter, dialyzed against ultrapure water with a 1 kDa MWCO before lyophilization.

### Reporter-independent glycan measurements

Total levan content was assayed using the Fructan Assay Kit (K-FRUC, Megazyme), modified to include the hydrolysis and absorbance measurement at 410 nm against a standard curve of 0.2–1 mg levan. CS abundance was estimated using the Total Glycosaminoglycan Assay Kit (K2085, BioVision) according to the manufacturer's directions against a standard curve of 2–10 μg CS.

### Statistics and reproducibility

No statistical method was used to predetermine sample size. Sample sizes were chosen based on the limits of the instrumentation to simultaneously measure multiple reactions in 384-well plates. No data were excluded from the analyses. The experiments were not randomized. The Investigators were not blinded to allocation during experiments and outcome assessment. All experiments were independently repeated at least twice. Tecan iControl 2.0 and Microsoft Excel 365 were used to collect and compute absorbance and luminescence data. Repeated measurements were analyzed by paired, two-tailed student's t-test, 1-way, or 2-way ANOVA using Graphpad Prism 9.3.1 where appropriate as indicated in each figure legend.

### Reporting summary

Further information on research design is available in the Nature Portfolio Reporting Summary linked to this article.

## Data availability

Source data are provided with this paper.

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

## Acknowledgements
We would like to thank the Eduardo Groisman and Mauricio Pontes labs for providing strains, reagents, and constructive feedback. We thank the Andy Goodman Lab and Bentley Lim for gifting plasmid, pSIE1, and helpful discussions. This work was supported by NIAID AI149319 awarded to G.E.T. and start-up funds from Penn State College of Medicine to G.E.T.

## Author contributions
J.L.M. and V.H.P. conducted experiments. J.L.M. and G.E.T. conceived and designed the research, constructed strains and plasmids, analyzed data, and wrote the manuscript. All co-authors contributed the final edited manuscript.

## Competing interests
The authors declare no competing interests.
