## [Peer Review File · Nature Communications]

REVIEWER COMMENTS

Reviewer #1 (Remarks to the Author):

This manuscript by the Townsend group presents a nifty tool employing luminescent gut microbes (*B. theta*) as sensors for transcriptional activity on polysaccharide utilization loci (PULs). The described tool (pBolux plasmid) harnesses the innate sensors and surface glycan binding proteins found on bacteria, and it is sensitive – it detects low amounts of soluble glycans, and it is selective – it can discriminate among structurally similar glycans. The tool will accelerate and facilitate new biological discoveries for how gut microbes utilize glycans as nutritional sources (and they showcase an example), and it will be of wide interest to many in the microbiome and glycobiology fields.

Major:

- A plot from Fig 6 showing correlation of actual transcriptional activity (mRNA transcript level) and luminescence from pBolux would significantly enhance the manuscript. Does the luminescence reporter faithfully recapitulate qPCR responses in dynamic range and kinetic activity?
- Could the authors comment on how much pBolux expression levels are necessary for efficient reporter activity? Does over-expression at all affect/compete for endogenous *susC*, *susD* activities?

Minor:

- Some editing to make the manuscript more concise would be appreciated. Currently, the manuscript is cumbersome to read with all the similar abbreviations (BTXXX, *susXXX*) that are not intuitive. There are extraneous information (X-fold serial dilutions, etc.) that hamper reading. Could the authors add descriptors in the figures or relegate some controls into the SI?
- Some clarification regarding heparin vs heparan sulfate is necessary. Heparin is the more sulfated and shorter cousin of heparan sulfate.
- Please indicate the degree of polymerization for CS used from Sigma.
- Figure 1 caption, what is IM?
- Fig 7 caption, the authors may have confused the open and filled bars in D and E. The filled bars refer to BT1763 and the open bars refer to BT3332.
- Molecular weight markers in Figure 7A would be helpful on the gel lanes.

Reviewer #2 (Remarks to the Author):

In this article by Modesto et al they describe the development of a glycan sensor system incorporating polysaccharide utilization loci (PUL) promoters with a luciferase reporter system. The article itself is a method development. While there are interesting observations about the use of specific glycans in certain *Bacteroides* sp, these observations aren't necessarily of great importance to our understanding of bacterial physiology or pathophysiology. With that said - the method they have developed and the importance of the method to the study of glycobiology is relevant and I believe in itself worthy of publication. It would be nice to see the application of this method in more complex systems (I.e. biologic samples with complex unknown glycan mixtures) or for validation of PUL with unknown glycan substrates as they themselves discuss. These would be true validation steps for this method. The building of a luciferase reporter system is in itself not novel as they reference other systems in *Bacteroides*. The use of the reporter system for the specific study of glycans is relevant and I think important to understand the importance of considering the difficulties of working with individual glycans in vitro.

I don't have major concerns with the methods. The experiments and controls are well done and support the claims in the manuscript.

Minor Concern

- As it stands the method is relevant only to the science of glycobiology and has applications in the study of a specific glycan in a specific microbe (which is meaningful). With that said it is a relevant contribution to the field and one that I think will be utilized moving forward. To improve the impact of the article I might suggest testing the biologic relevance of the system by using a more complex glycan mixture, ideally a biologic sample of unknown glycan composition but with the target glycan. It would be interesting to compare how this reporter predicts glycan utilization vs commercial assays as they have done in other experiments. I just think this experiment would be a step towards validation for their discussion which as it stands I believe is relevant and a natural extension from the controlled experiments they presented.

Dear Reviewers,

We would like to thank you for the positive remarks and useful suggestions regarding our manuscript entitled, "*Harnessing gut microbes for glycan detection and quantification*". Attached please find our revised manuscript in which we incorporated your feedback and added new experimental evidence that strengthen our findings. Below, please find our point-by-point responses to your specific comments.

Reviewer #1

A plot from Fig 6 showing correlation of actual transcriptional activity (mRNA transcript level) and luminescence from pBolux would significantly enhance the manuscript. Does the luminescence reporter faithfully recapitulate qPCR responses in dynamic range and kinetic activity?

We now demonstrate that *luxC* transcript levels and the corresponding luminescence follow similar trends that change over time indicating that the concentration dependent increases in luminescence are the product of PUL-sensor driven gene transcription in response to target glycan availability as shown in Supplementary Fig. 8. We have amended the text to describe these results in lines 317-320, which support our use of area under the curve to quantify reporter responses.

Could the authors comment on how much pBolux expression levels are necessary for efficient reporter activity?

We now demonstrate that expression from the *rpoD* promoter in a single copy plasmid, which is 10-fold lower than the multi-copy *pBolux* plasmid, is sufficient to detect luciferase activity over a vector control in Supplementary Fig. 3b. We have amended the text to describe these results in lines 151-154.

*Does over-expression at all affect/compete for endogenous *susC*, *susD* activities?*

We compared *susC* levels in *wild-type* strains with and without the reporter plasmid. We find that although the *susC* transcript levels are lower in the reporter strain, as shown in Supplementary Figs. 4c and 5d, this is negligible to target glycan utilization, which requires *susC* activity and is not impaired by the presence of the reporter plasmid. We have amended the text to describe these results in lines 175-178 for P-BT3332 and lines 222-225 for P-BT1763.

Some editing to make the manuscript more concise would be appreciated. Currently, the manuscript is cumbersome to read with all the similar abbreviations (BTXXX, susXXX) that are

not intuitive. There are extraneous information (X-fold serial dilutions, etc.) that hamper reading. Could the authors add descriptors in the figures or relegate some controls into the SI?

We have extensively edited the manuscript to eliminate repetitive and cumbersome genetic annotations where possible and modified the figures to clarify the experiments in each panel. Furthermore, 8 panels from the main text figures have been relocated to the supplementary information to reduce the size of the main figures and improve clarity.

Some clarification regarding heparin vs heparan sulfate is necessary. Heparin is the more sulfated and shorter cousin of heparan sulfate.

We regret this error. We used heparin, which has been corrected in the main text, figures, and figure legends. Additionally, we have included all glycan manufacturer and catalog numbers to improve clarity.

Please indicate the degree of polymerization for CS used from Sigma.

We have amended the text to reflect that the CS used here is highly polymeric and have provided vendors and catalog numbers for all glycans examined in this study. In this instance, we used CS-A from bovine trachea (MilliporeSigma, C9819), which contains CS polymers ranging from 20-26 kDa in size and are approximately 80-120 monosaccharide units in length.

Figure 1 caption, what is IM?

We have updated the Figure 1 legend to indicate that IM refers to the bacterial inner membrane.

Molecular weight markers in Figure 7A would be helpful on the gel lanes.

We have repeated the affinity-PAGE experiments with NativeMark (ThermoFisher) molecular weight markers and replaced panels a-c in Figure 7 with these images. All images are available in the Supplementary Information.

Reviewer #2

As it stands the method is relevant only to the science of glycobiology and has applications in the study of a specific glycan in a specific microbe (which is meaningful). With that said it is a relevant contribution to the field and one that I think will be utilized moving forward. To improve the impact of the article I might suggest testing the biologic relevance of the system by using a more complex glycan mixture, ideally a biologic sample of unknown glycan composition but with the target glycan. It would be interesting to compare how this reporter predicts glycan utilization vs commercial assays as they have done in other experiments. I just think this experiment would be a step towards validation for their discussion which as it stands I believe is relevant and a natural extension from the controlled experiments they presented.

We have performed several new experiments to address all of Reviewer 2's suggestions to enhance the impact of our study by applying our method "using a more complex glycan mixture, ideally a biologic sample of unknown glycan composition but with the target glycan." We measured the abundance of CS/HA in a biologically derived glycan mixture prepared from the porcine mucosa, called PMOG, using our reporter system. Furthermore, we constructed a new reporter plasmid from a PUL that targets an unknown glycan present in PMOGs, P-BT2818, and demonstrate that a strain harboring this plasmid exhibits increased bioluminescence following the introduction of PMOG, which requires its linked sensor, BT2826 (Fig. 7g). Finally, we show

that the CS-specific SGBP, BT3330, can retain target glycans from PMOG and this co-purifying material is unable to activate P-*BT2818* indicating enrichment for target molecules. These new experiments demonstrate the utility of our reporter system for: 1.) detection of known target glycans in biologically-derived mixtures of unknown composition (Fig. 7f), 2.) construction of a reporter plasmid for a PUL with an unknown target glycan (Fig. 7g), 3.) implementation of PUL reporters to detect unknown glycans in undefined biologically derived mixtures (Fig. 7g), and 4.) the implementation of PUL reporters to determine enrichment of target glycans from undefined biologically derived mixtures using SGBPs (Fig. 7i and Supplementary Fig. 11c). This additional work is described in detail in lines 463-497.

Finally, during the production of our re-submission, we discovered errors in the calculation and presentation of CS and levan concentrations. Those values have been corrected, the raw data is available in the Supplementary Information, and are now appropriately reported in this revised manuscript. Importantly, these changes do not alter the conclusions of the manuscript.

We believe that all comments and suggestions made by the reviewers have been addressed. We hope you will find our revised manuscript suitable for publication in Nature Communications. Thank you for your consideration of the revised manuscript.

Sincerely,

Guy E. Townsend II

REVIEWERS' COMMENTS

Reviewer #1 (Remarks to the Author):

The authors have sufficiently addressed my concerns, and I recommend publication in Nat Comms.

Reviewer #2 (Remarks to the Author):

The authors have addressed my concerns and did a great job demonstrating the utility of this system in complex mixtures and for unknown glycans.